# Effects of Melatonin Supplementation on Lipid Metabolism and Body Fat Accumulation in Ovariectomized Rats

**DOI:** 10.3390/nu15122800

**Published:** 2023-06-19

**Authors:** Ling-Wen Hsu, Yi-Wen Chien

**Affiliations:** 1Department of Nutrition and Health Science, Taipei Medical University, Taipei 11031, Taiwan; ma07110020@tmu.edu.tw; 2Research Center of Geriatric Nutrition, College of Nutrition, Taipei Medical University, Taipei 11031, Taiwan; 3Graduate Institute of Metabolism and Obesity Sciences, Taipei Medical University, Taipei 11031, Taiwan; 4Nutrition Research Center, Taipei Medical University Hospital, Taipei 11031, Taiwan; 5TMU Research Center for Digestive Medicine, Taipei Medical University, Taipei 11031, Taiwan

**Keywords:** melatonin, postmenopausal obesity, ovariectomy, lipid metabolism, body fat accumulation

## Abstract

Postmenopausal obesity is a rising problem. Melatonin (Mel) is a hormone secreted by the pineal gland that regulates the circadian rhythms and improves obesity. In this experiment, ovariectomized (OVX) rats were used as a menopause model to explore the effects of Mel supplementation on lipid metabolism, body fat accumulation, and obesity. Nine-week-old female rats underwent an OVX surgery and were assigned to the following groups: control group (C), low-dose group (L, 10 mg/kg body weight (BW) Mel), medium-dose group (M, 20 mg/kg BW Mel), and high-dose group (H, 50 mg/kg BW Mel), administered by gavage for 8 weeks. The results showed that the OVX rats supplemented with low, medium, and high doses of Mel for 8 weeks exhibited reduced BW gain, perirenal fat mass, and gonads fat mass, and an increased serum irisin level. Low and high doses of Mel induced brite/beige adipocytes in the white adipose tissues. In addition, the messenger RNA levels of the fatty acid synthesis enzymes were significantly reduced after the high-dose Mel supplementation. Thus, Mel can reduce the hepatic fatty acid synthesis and promote the browning of white adipose tissues through irisin; thereby, improving obesity and body fat accumulation in OVX rats.

## 1. Introduction

Obesity is a global health issue characterized by greater energy intake than energy expenditure, with excess energy stored as triglycerides (TGs) in white adipose tissues (WATs), resulting in fat accumulation. In addition, obesity-related complications include dyslipidemia, hypertension, non-alcoholic fatty liver, type 2 diabetes, heart disease, and some types of cancers [1,2], which increase the burden of social medical care. Therefore, it is important to find treatments and interventions for obesity. 

Menopause is the spontaneous cessation of menstrual cycles for one year and is caused by the loss of ovarian function, usually around 45–55 years old [3]. Postmenopausal women have a higher incidence of body fat accumulation and obesity than premenopausal women. Decreased estrogen secretion in postmenopausal women induces changes in the body fat distribution, especially an increased visceral fat accumulation, leading to abdominal obesity [4,5,6,7]. Abdominal fat has a high rate of lipolysis, which triggers excess free fatty acid (FFA) production from excessive visceral fat lipolysis, promotes insulin resistance (IR), and leads to metabolic diseases [8]. An ovariectomy (OVX), surgery to remove the ovaries to emulate a state of low estrogen, is a common animal model of menopause [9]. An estrogen deficiency induced by an OVX in rats results in moderate excess weight gain [10,11] and fat accumulation in the liver or WAT. Previous studies showed that the expressions of genes related to β-oxidation (such as carnitine palmitoyl transferase I (CPT-1)) and fatty acid synthesis (such as acetyl-CoA carboxylase (ACC)) respectively decreased and increased in OVX rodents, resulting in hepatic fat accumulation [12,13,14,15]. Furthermore, other studies showed that menopausal women had lower lipolysis rates and higher lipoprotein lipase (LPL) activity in adipose tissue, which might contribute to body fat accumulation [16].

Irisin is a myokine, which is produced by the cleavage of fibronectin type III domain-containing 5 (FNDC5) in muscles. Irisin can stimulate the browning of WATs by increasing the uncoupling protein-1 (UCP-1) inducing non-shivering thermogenesis, thereby, increasing the energy expenditure [17,18]. A previous study showed that OVX rats exhibited lower irisin levels and higher body weight (BW) gain, which recovered to a level comparable to a control group with continuous irisin treatment, thereby, reducing weight gain. [19]. 

Melatonin (Mel; *N*-acetyl-5-methoxytryptamine) is mostly provided by the pineal gland [20]. Mel is responsible for regulating circadian rhythms [21,22,23], reducing inflammatory responses [24,25,26] and oxidative stress [26,27,28], and as an anti-obesity agent [29]. Several studies showed that Mel supplementation suppressed BW gain in obese rodents without affecting food intake [30,31,32,33]. Another study suggested that Mel could regulate lipid metabolism, as male rats treated with 10 mg/kg BW Mel for 60 days had significantly reduced the lipogenesis-related gene expression, and promoted the lipolysis-related gene expression, and significantly reduced the hepatic lipid and plasma lipid indices [34]. In addition, some physiological or pathophysiological states, such as aging [35], shift work [36], and nocturnal light pollution [37,38], can lead to lower circulating Mel levels, which are accompanied by an energy metabolism imbalance, IR, obesity, and metabolic syndrome. A clinical study showed that postmenopausal women had significantly lower Mel levels than premenopausal women [39,40]. However, the effects and mechanisms of Mel on improving obesity in OVX rats are still unclear.

As a result, this study aimed to explore the effects and underlying mechanisms of Mel on lipid metabolism, body fat accumulation, and obesity in OVX rats.

## 2. Materials and Methods

### 2.1. Animals

Twenty-four female Sprague–Darley rats that underwent an OVX (9 weeks of age) were purchased from BioLasco (Taipei, Taiwan). The experimental rats were housed in an animal room with a 12 h light/dark cycle (lights on at 07:00). Additionally, the temperature and relative humidity were controlled at 22 ± 2 °C, 65% ± 5%. The experiment started after 4 weeks of the surgical recovery period. During the surgical recovery period, water and the standard laboratory diet (Rodent Laboratory Chow 5001; PMI Nutrition International, St. Louis, MO, USA) were allowed ad libitum. This experiment was approved by the Institutional Animal Care and Use Committee of Taipei Medical University (IACUC no.: LAC-2020-0355).

### 2.2. Experimental Design

All OVX rats were randomly assigned to different experimental groups (*n* = 6 in each group) after a surgical recovery period, including a control (C, received the 8% ethanol vehicle) group, low-dose (L, 10 mg Mel/kg BW/day), medium-dose (M, 20 mg Mel/kg BW/day), and high-dose (H, 50 mg Mel/kg BW/day) groups. Mel (Sigma-Aldrich, St Louis, MO, USA) was dissolved in an 8% ethanol (*w*/*v*) aqueous solution and given by gavage (at 18:00–19:00). During the 8 weeks of the Mel intervention, BW, food intake, and drinking water were recorded weekly, and the feed efficiency ratio (FER, %; food intake/BW gain) was calculated. Rats were sacrificed with a mixed solution of zoletil and rompun (1 mL/kg BW, intraperitoneal injection) after 8 weeks of the experiment. Serum was collected and centrifuged at 3500× *g* at 4 °C for 15 min. The uterus, liver, gonadal WATs, perirenal WATs, quadricep femur muscle tissues, and gastrocnemius muscle tissues were dissected and weighed. All samples were stored at −80 °C for further analysis.

### 2.3. Serum Measurements

The serum estradiol and follicular-stimulating hormone (FSH) level were analyzed by an enzyme-linked immunosorbent assay (ELISA) Kit (Wuhan Fine Biotech, Wuhan, China). The fasting glucose level was tested by a glucose monitor (Abbott Diabetes Care, Oyl, Witney, UK); the serum insulin level was analyzed using an ELISA kit (Mercodia, Uppsala, Sweden); and the homeostasis model assessment (HOMA)-IR index was calculated by applying the equation: HOMA-IR = (fasting blood glucose (mg/dL) × insulin (mIU/L))/405. The serum leptin concentration was measured using an ELISA Kit (BioVender, Brno, Czech Republic); the serum adiponectin concentration was analyzed using an ELISA Kit (AssayPro, St. Charles, MI, USA); and the serum irisin level was analyzed with an ELISA Kit (Bio Vender, Brno, Czech Republic).

### 2.4. Hepatic Lipid Measurements

Total hepatic lipid was extracted via the Folch methods [41]; the hepatic TC and TG contents (milligrams per gram (mg/g) of liver tissue) were analyzed by a colorimetric assay kit (Randox Laboratories). 

### 2.5. Real-Time Reverse-Transcription Polymerase Chain Reaction (RT-PCR)

Total RNA from the liver, gonadal adipose tissues, and quadricep femur muscle tissues were extracted with a Trizol reagent (Life Technologies, Carlsbad, CA, USA). The messenger (m)RNA levels of ACC, FAS, CPT-1, acyl CoA oxidase (ACO), peroxisome proliferator-activated receptor alpha (PPARα), 5′AMP-activated protein kinase (AMPK), LPL, hormone-sensitive lipase (HSL), PPARγ, fibronectin type III domain-containing protein 5 (FNDC5), and peroxisome proliferator-activated receptor-gamma coactivator (PGC-1α) were quantified by a real-time qPCR following previous methods [42]. Data were normalized using β-actin as an internal control, and the relative gene expression was calculated using the 2^−ΔΔCt^ method. The primer sequences are shown in Table 1.

### 2.6. Histopathological Examination

Liver and gonadal WATs were fixed in 10% neutral buffered formaldehyde. Samples were trimmed, embedded in paraffin, sectioned, stained with hematoxylin and eosin (H&E), and examined microscopically by a veterinarian pathologist who was blinded to the nature of the treatment received by the respective groups. Histopathological evaluations were graded as described by Shackelford et al. and Kleiner et al. [43,44].

### 2.7. Statistical Analysis

Data were presented as the mean ± standard error of the mean (SEM) and analyzed by GraphPad Prism vers. 8.0 software (GraphPad Software, San Diego, CA, USA). During the Mel intervention period, the differences among all groups were compared using a one-way analysis of variance (ANOVA) followed by the Tukey’s test. Statistical significance was accepted at *p* < 0.05.

## 3. Results

### 3.1. Female Hormone-Related Variables in OVX Rats

In this experiment, OVX was used to simulate the decline in estrogen concentration in postmenopausal women. Female hormone-related variables and uterus weight were measured to ensure the effect of the OVX. As shown in Table 2, 10 days after the OVX, no difference was found in the serum estradiol or FSH levels among all the groups. After 8 weeks of the intervention, the uterus weights showed no differences among all the groups.

### 3.2. Effects of Mel on BW and Food Intake Variables in OVX Rats

At the baseline and after the recovery period, a significant difference was not observed in BW among all the groups. After 8 weeks of the Mel intervention, no difference was found in BW among all the groups. Total BW gain was significantly decreased in the Mel treatment groups (L, M, and H groups) compared to the C group, but significant differences were not observed among the Mel treatment groups. There were no significant differences in the food intake among all the group. Compared to the C group, the food efficiency was significantly lower in the L and M groups, and a decreasing trend was shown in the H group (*p* = 0.057) (Table 3).

### 3.3. Effects of Mel on Organ, Adipose Tissue, and Muscle Tissue Mass in OVX Rats

After 8 weeks of the Mel intervention, the gonadal and perirenal adipose tissue weights were significantly lower in the Mel treatment groups than in the C group, but there were no significant differences among the Mel treatment groups. There were no significant differences in the liver weight, quadricep femoris muscle weight or gastrocnemius muscle weight among all the groups. In terms of the relative organ or tissue mass, compared to the C group, the relative gonadal adipose tissue weight was significantly lower in the H group. Moreover, compared to the C group, the relative perirenal adipose tissue weight showed a decreasing trend in the Mel treatment groups, especially in the H group (C vs. H *p* = 0.08) (Table 4). 

### 3.4. Effects of Mel on IR in OVX Rats

After 8 weeks of the Mel intervention, there was no significant difference in the serum insulin concentration among all the groups. In addition, no changes were seen in the fasting glucose or HOMA-IR among all the groups; only a declining trend was observed in the Mel treatment groups compared to the C group (Table 5).

### 3.5. Effects of Mel on Adipokine in OVX Rats

For adipokine concentrations, the serum leptin and adiponectin levels were analyzed. After 8 weeks of the Mel intervention, the Mel treatment groups exhibited slightly lower serum leptin levels compared to the C group, but the differences did not reach statistical significance. In addition, there was no significant difference in the serum adiponectin levels among all the groups (Figure 1).

### 3.6. Effects of Mel on Hepatic Lipid Metabolism in OVX Rats

#### 3.6.1. Hepatic Lipid Contents

To determine whether Mel could attenuate the hepatic lipid accumulation, the hepatic TC and TG contents were examined, as shown in Figure 2. After 8 weeks of the Mel intervention, there were no significant differences in the hepatic TC or TG levels among all the groups. 

#### 3.6.2. Hepatic Lipid Metabolism-Related mRNA Levels

As shown in Figure 3, for the fatty acid synthesis pathway, ACC and FAS mRNA levels in the liver were significantly lower in the H group than in the C group, and a slight decrease was seen in the L and M groups. For the fatty acid oxidation pathway, there were no significant differences in the CPT-1 or ACO mRNA levels in the liver among all the groups. In terms of the mRNA levels of important transcriptional regulators in the hepatic lipid metabolism, such as PPAR-α or AMPK, there were no significant differences among all the groups.

#### 3.6.3. Hepatic Histopathology

According to the pathological histologic images of the liver, steatotic and necrotic lesions were not observed in any of the groups. Focal inflammatory cell infiltration in the hepatocytes was observed in all the groups (Figure 4). In terms of the total histopathological scores, no statistical difference was observed in the lesions scores among all the groups (Table 6). 

### 3.7. Effects of Mel on Irisin and Irisin-Related Levels in OVX Rats

Irisin, the cleaved form of the FNDC5 protein, acts as a linkage between the muscles and WATs. PGC-1α and FNDC5 mRNA levels in the quadricep femur muscle tissues and in serum irisin concentration are shown in Figure 5. PGC-1α and FNDC5 mRNA levels showed no differences among all the groups. Compared to the C group, the serum irisin levels were significantly higher in the Mel treatment groups, but there were no significant differences among the Mel treatment groups.

### 3.8. Effects of Mel on WAT Morphology in OVX Rats

According to the pathological histologic images of the gonadal adipose tissues after H&E staining, brite/beige adipocytes were found in the L and H groups, but no brite/beige adipocytes were observed in the C or M groups (Figure 6). In addition, there were no significant differences in the scores of brite/beige adipocytes among all the groups (Table 7).

### 3.9. Effects of Mel on Adipose Tissue Lipid Metabolism-Related Enzymes mRNA Levels in OVX Rats

LPL delivers fatty acid into the adipose tissue, and HSL is an important enzyme in the lipolysis process. As a result, to evaluate the effect of Mel on lipid metabolism in the adipose tissue, the mRNA levels of LPL and HSL were analyzed. After 8 weeks of the Mel intervention, no significant differences were found in the LPL and HSL mRNA levels among all the groups. PPAR-γ is a major regulator of adipogenesis, and PGC-1α is a pivotal transcriptional factor that regulates thermogenesis and browning of WAT. After 8 weeks of the Mel intervention, no changes were observed in the PPAR-γ and PGC-1α mRNA levels (Figure 7). 

## 4. Discussion

In this study, we used the same study design of OVX rats as our laboratory’s previous study [45] to mimic the decrease in the estrogen levels during menopause. Yeh et al. found that compared to sham-operated rats, OVX rats had significantly lower estrogen levels and uterine atrophy, accompanied by increasing BW gain and abdominal fat accumulation, which was consistent with other previous studies [12,46,47]. In addition, Yeh et al. showed that the serum estradiol level in the sham group was 57.52 ± 7.09 (ng/mL), and the weight of the uterus was 0.56 ± 0.08 (g), which were much higher than the OVX rats in this experiment, and there were no significant differences among all the groups. Therefore, it was speculated that the ovariectomy in this experiment was successful. Estrogen plays a crucial role in regulating energy metabolism. Estrogen can reduce appetite by acting directly on anorexigenic and orexigenic neurons in the hypothalamus. In addition, it can also increase thermogenesis in brown adipose tissues (BATs) through the ventromedial hypothalamus-sympathetic nervous system (VMH-SNS)-BAT pathway. As a result, obesity caused by OVX may be attributed to increased food intake and reduced energy expenditure due to an estrogen deficiency [48].

In the present study, OVX rats treated with Mel supplementation reduced BW gain without affecting food intake, but no dose-dependent effect was found. Similar to previous studies, exogenous Mel supplementation could prevent weight gain in male mice [34], especially in mice with high-fat diet (HFD)-induced obesity [49,50], and no dose effect was found [51]. A meta-analysis clinical study indicated that compared to a placebo, Mel supplementation reduced BW, and the results were better in studies using doses of ≤8 mg/day. This means that the effect of Mel on BW is not dose dependent. Moreover, the present study found that the abdominal visceral fat mass, such as the gonadal and perirenal fat masses, decreased in OVX rats with Mel supplementation. Similar to previous studies, such as that by Majumdar et al. who found that the intra-abdominal fat mass was decreased in OVX rats with fructose diet-induced obesity and 3 mg/kg BW Mel supplementation in drinking water [52]. Another clinical study showed that the fat mass decreased in postmenopausal women with 1 or 3 mg/day Mel supplementation for 1 year [53]. Thus, melatonin supplementation ameliorated body weight and abdominal visceral fat accumulation in OVX rats.

In terms of adipokines, although there were no significant differences in the serum leptin or adiponectin levels among all the groups, compared to the C group, the Mel treatment groups showed a trend of lower serum leptin levels. Previous studies showed that melatonin reduced the serum leptin level [30,35], and another study indicated that the reduced leptin level after melatonin treatment was associated with reducing fat mass rather than improving the leptin function [54]. The result of the leptin level in this study was consistent with the Szewczyk-Golec et al. study, as male rats treated with 20μg/mL Mel for 7 weeks had significantly reduced body weight, but there was no significant difference in the leptin level [54]. Leptin and melatonin are secreted in a circadian rhythm with acrophases in the nocturnal period. Therefore, melatonin did not affect the leptin level in this study, which might be related to the different time of sample collection, because of the rhythmic secretion of leptin. However, a study by Yeh et al. pointed out that compared with the sham group, there was no significant difference in the leptin level in OVX rats [45]. Consistent with the study by Babaei et al., eight weeks after ovariectomy, the body weight and visceral fat mass were significantly increased in 11-week-old OVX rats but did not affect the adiponectin level. The possible reason is that OVX can promote adipocytes hyperplasia, not adipocyte hypertrophy; only adipocyte hypertrophy will reduce adiponectin level [55]. Therefore, we speculate that the OVX animal model in this study did not cause adiponectin secretion disorder. If the secretion is normal, the intervention of melatonin will not affect the adiponectin level. 

In terms of hepatic lipid metabolism, this study demonstrated that the mRNA levels of the hepatic fatty acid synthesis enzymes, ACC and FAS, were significantly lower in the OVX rats treated with 50 mg/kg BW Mel for 8 weeks, but no difference was found in the hepatic fatty acid oxidation enzymes. Past studies found that Mel could regulate the hepatic lipid metabolism. Ou et al. administered 10, 20, and 50 mg/kg BW of Mel to hamsters with HFD-induced hyperlipidemia, and found that Mel treatment decreased the activities of the ACC and FAS lipogenic enzymes, and increased the mRNA levels of the CPT-1 lipolysis enzymes, while reducing the hepatic TC and TG contents, thereby, decreasing serum TC, TG, and low-density lipoprotein cholesterol (LDL-C) levels [56]. Liu et al. reported that a daily subcutaneous injection of 10 mg/kg BW Mel activated the liver AMPKα/PPARα/CPT-1 pathway in guinea pigs with glucose and lipid metabolism disorders caused by persistent artificial light exposure and an HFD. Generally, AMPKα can regulate PPARα expression. After its activation by AMPKα, PPARα reduces TG synthesis in the liver by enhancing the CPT-1 expression [57]. It was speculated that the therapeutic effect of Mel on hepatic fatty acid oxidation was inconsistent with past studies, and possible reasons were as follows: Firstly, compared to an HFD-induced obesity animal model, OVX rats had a milder degree of obesity and liver lipid accumulation, so the effect of Mel regulating the fatty acid oxidation pathway was less significant, resulting in no change in the hepatic lipid contents. This explanation can be verified from the hepatic histopathology results, as steatotic and necrotic lesions were not observed in any of the groups. Secondly, it might be related to the method of the Mel intervention. Compared to a subcutaneous injection of Mel, the absorption rate of oral Mel is lower, and the bioavailability is only 9%~33% [58], thus, the designed dosage in this study might not have been sufficient to affect the hepatic lipid metabolism pathways. On the other hand, Heo et al. reported that the administration of 50 or 100 mg/kg BW of Mel for 10 weeks to rats with HFD-induced obesity resulted in hepatic histopathological images showing that 100 mg/kg BW Mel significantly reduced the hepatic steatosis scores, but 50 mg/kg BW Mel showed a downward trend, and the differences did not reach statistical significance [59]. As a result, by increasing the dosage of Mel or the intervention time, one may be able to observe more-significant effects of Mel on hepatic lipid metabolism.

Previous studies suggested that Mel had an anti-obesity effect by the browning of WATs, thereby increasing energy expenditure [51,60]. Obese Zücker diabetic fatty (ZDF) rats were supplemented with 10 mg/kg Mel in drinking water for 6 weeks, and the number of brite/beige adipocytes in inguinal WATs increased, and PGC-1α and UCP-1 levels were significantly higher and promoted non-shivering thermogenesis [60]. In another study, Tung et al. treated obesity rats with 10, 20, and 50 mg/kg BW Mel for 8 weeks, and found that Mel reduced BW gain and increased the circulating irisin concentrations; an increased number of brite/beige adipocytes was observed in the inguinal WATs [51]. In the present study, Mel supplementation elevated the serum irisin levels. In addition, 10 and 50 mg/kg BW Mel induced brite/beige adipocytes in the gonadal WATs, but no brite/beige adipocytes were observed in the M group. The possible reasons were as follows: The white adipose tissues for morphological analysis were randomly collected at the time of sacrifice. Additionally, the number of samples for morphological analysis was small (*n* = 3). Thus, the presence of brite/beige adipocytes was not observed in the M group. Moreover, compared to the study by Jiménez-Aranda et al., where they dissected the adipose tissue after undergoing acute cold exposure [60], this experimental design was more accurate and easier to find browning in the WATs. In future studies, acute cold exposure before sacrifice should be designed and it may be able to observe more-significant effects of melatonin on WAT browning. Previous studies suggested that Mel improved obesity possibly by being implicated in increased physical activity [30,33]. Exercise induces the skeletal muscles to cleave FNDC5, and irisin is produced. Once irisin is released into the circulation, it acts on white adipocytes to induce the browning response and subsequently activates non-shivering thermogenesis. Therefore, it was speculated that Mel supplementation could promote irisin secretion, which might be related to physical activity. Although the FNDC5 mRNA levels in muscle tissues only exhibited an upward trend in the melatonin treatment group, the serum irisin concentration was significantly increased in the melatonin treatment group. In addition, the FNDC5 gene expression in the adipose tissue was lower than in the muscle tissue but also correlated with the serum irisin levels [61]. Thus, the expressions of FNDC5 proteins in the muscle or FNDC5 mRNA levels in the adipose tissue must be measured in future studies to ensure the explicit mechanism of irisin increase. However, some studies showed that Mel did not affect physical activity [62,63]. Jiménez-Aranda et al. indicated that Mel treatment did not affect circulating irisin concentrations or physical activity [60]. Whether Mel increases physical activity is still inconclusive. Although many studies found that Mel or irisin can promote the browning of adipose tissues, few studies had explored the relationship between Mel and irisin. Therefore, elucidating the exact mechanism by which Mel increases FNDC5/Irisin requires further animal and cell studies. However, the results of our study still showed that Mel may be beneficial for regulating the irisin levels and browning of WATs in OVX rats.

## 5. Conclusions

Mel can reduce the hepatic fatty acid synthesis pathway and promote browning of WATs through irisin, thereby improving obesity and body fat accumulation in OVX rats. Thus, Mel supplementation can possibly be a helpful weight management strategy for postmenopausal women.

## Figures and Tables

**Figure 1 nutrients-15-02800-f001:**
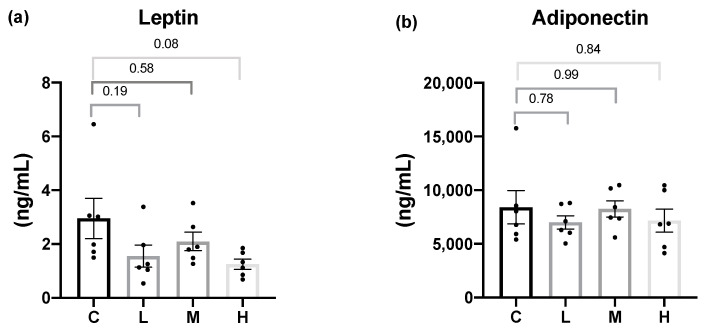
Serum adipokine concentrations after 8 weeks of the melatonin intervention. (**a**) Serum leptin levels; (**b**) serum adiponectin levels. All values are presented as the mean ± standard error of the mean (*n* = 6). Differences between different groups were determined by a one-way ANOVA with Tukey’s test. Black dots represented each sample. Abbreviations: C, control group (ovariectomized (OVX) rats); L, low-dose group (OVX rats treated with 10 mg/kg body weight (BW) melatonin); M, medium-dose group (OVX rats treated with 20 mg/kg BW melatonin); H, high-dose group (OVX rats treated with 50 mg/kg BW melatonin).

**Figure 2 nutrients-15-02800-f002:**
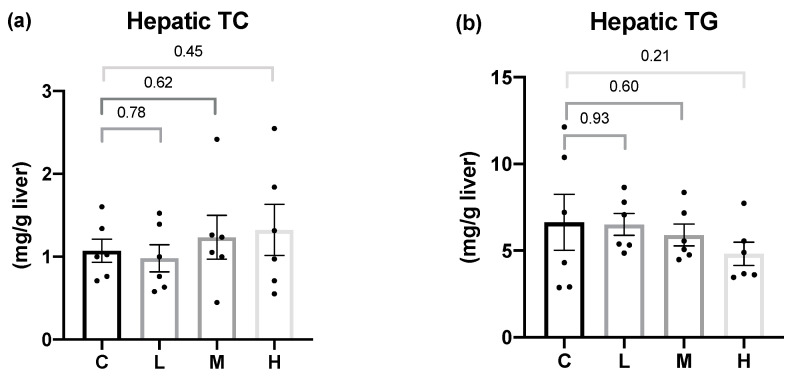
Hepatic lipid contents after 8 weeks of the melatonin intervention. (**a**) Hepatic total cholesterol (TC) levels; (**b**) hepatic triglyceride (TG) levels. All values are presented as the mean ± standard error of the mean (*n* = 6). Differences between different groups were determined by a one-way ANOVA with Tukey test. Black dots represented each sample. Abbreviations: C, control group (ovariectomized (OVX) rats); L, low-dose group (OVX rats treated with 10 mg/kg body weight (BW) melatonin); M, medium-dose group (OVX rats treated with 20 mg/kg BW melatonin); H, high-dose group (OVX rats treated with 50 mg/kg BW melatonin).

**Figure 3 nutrients-15-02800-f003:**
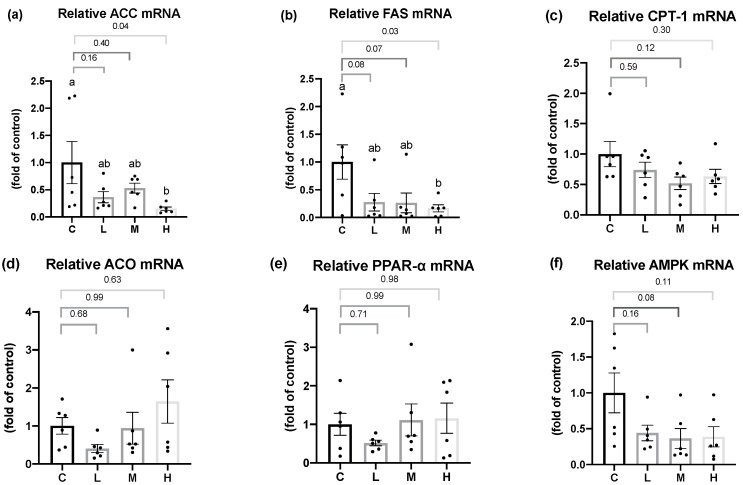
Hepatic lipid metabolism-related mRNA levels after 8 weeks of the melatonin intervention. Relative (**a**) acetyl-CoA-carboxylase (ACC), (**b**) fatty acid synthase (FAS), (**c**) carnitine palmitoyl transferase (CPT)-1, (**d**) acetyl-CoA oxidase (ACO), (**e**) peroxisome proliferator activated receptor α (PPARα), and (**f**) 5′AMP-activated protein kinase (AMPK) mRNA levels. All values are presented as the mean ± standard error of the mean (*n* = 6). Different superscripts (^a, b^) indicate a significant difference between different groups at *p* < 0.05 by a one-way ANOVA with Tukey test. Black dots represented each sample. Abbreviations: C, control group (ovariectomized (OVX) rats); L, low-dose group (OVX rats treated with 10 mg/kg body weight (BW) melatonin); M, medium-dose group (OVX rats treated with 20 mg/kg BW melatonin); H, high-dose group (OVX rats treated with 50 mg/kg BW melatonin).

**Figure 4 nutrients-15-02800-f004:**
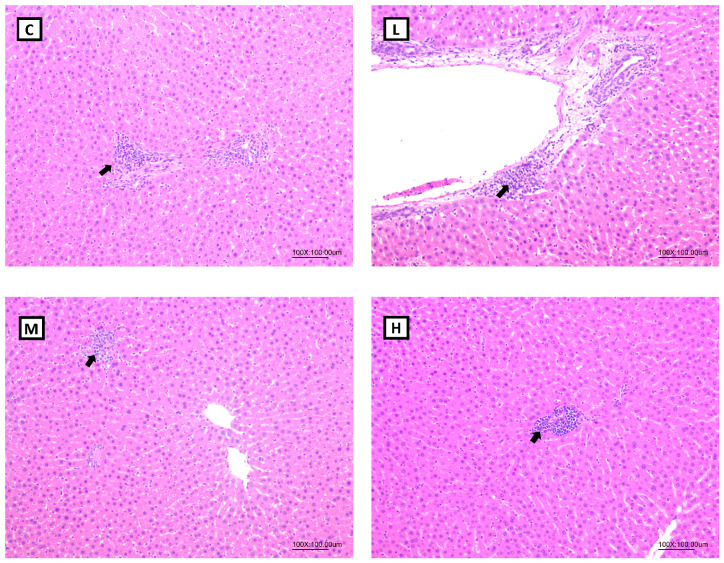
Hepatic histopathology of ovariectomized (OVX) rats after 8 weeks of a melatonin intervention (100×). Arrow: inflammatory cell infiltration. Abbreviations: C, control group (ovariectomized (OVX) rats); L, low-dose group (OVX rats treated with 10 mg/kg body weight (BW) melatonin); M, medium-dose group (OVX rats treated with 20 mg/kg BW melatonin); H, high-dose group (OVX rats treated with 50 mg/kg BW melatonin).

**Figure 5 nutrients-15-02800-f005:**
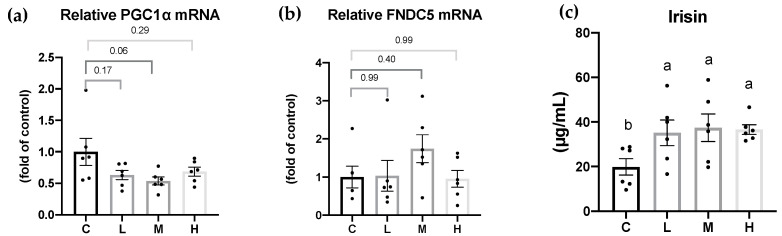
Peroxisome proliferator-activated receptor-gamma coactivator-1α (PGC-1α)/fibronectin type III domain-containing 5 (FNDC5) mRNA levels and serum irisin levels after 8 weeks of the melatonin intervention. Relative (**a**) FNDC5 mRNA levels in quadricep muscle tissues; and (**b**) serum irisin levels (**c**) serum irisin levels. All values are presented as the mean ± standard error of the mean (*n* = 6). Different superscripts (^a, b^) indicate a significant difference between different groups at *p* < 0.05 by a one-way ANOVA with Tukey test. Black dots represented each sample. Abbreviations: C, control group (ovariectomized (OVX) rats); L, low-dose group (OVX rats treated with 10 mg/kg body weight (BW) melatonin); M, medium-dose group (OVX rats treated with 20 mg/kg BW melatonin); H, high-dose group (OVX rats treated with 50 mg/kg BW melatonin).

**Figure 6 nutrients-15-02800-f006:**
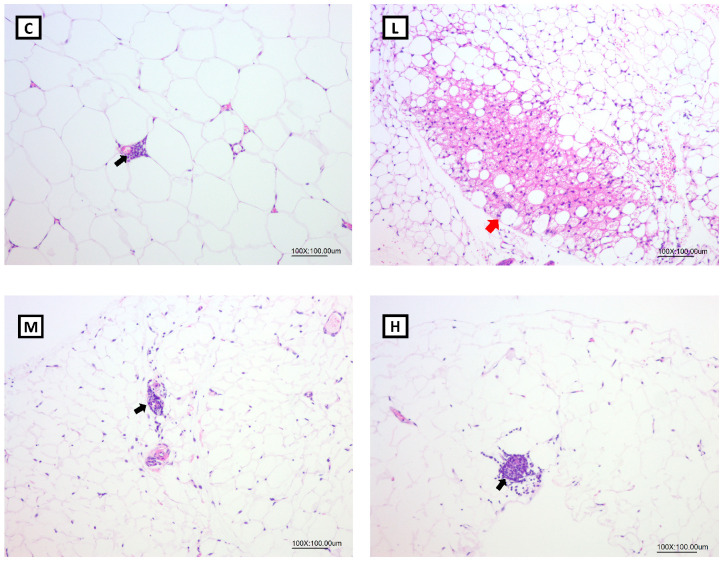
Pathological histologic images of gonadal adipose tissues after H&E staining (100×). Red arrow: brite/beige cell, black arrow: inflammation. All values are presented as the mean ± standard error of the mean (*n* = 3). Abbreviations: C, control group (ovariectomized (OVX) rats); L, low-dose group (OVX rats treated with 10 mg/kg body weight (BW) melatonin); M, medium-dose group (OVX rats treated with 20 mg/kg BW melatonin); H, high-dose group (OVX rats treated with 50 mg/kg BW melatonin).

**Figure 7 nutrients-15-02800-f007:**
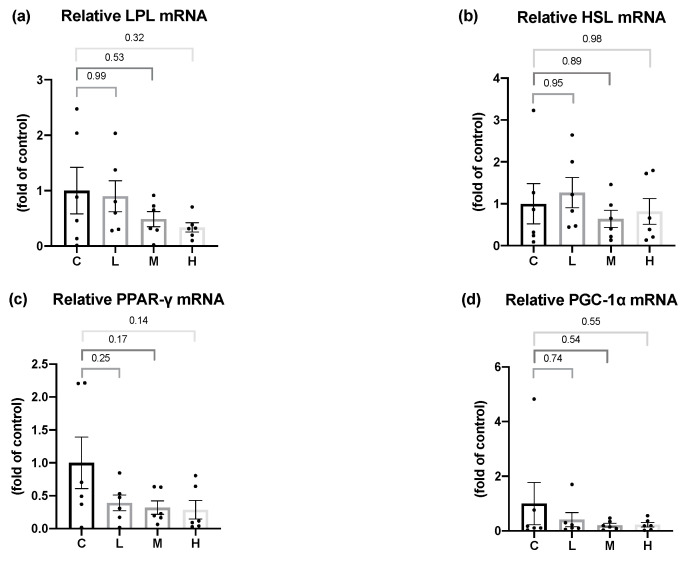
Lipid metabolism-related mRNA levels in gonadal white adipose tissues after 8 weeks of the melatonin intervention. Relative (**a**) lipoprotein lipase (LPL), (**b**) hormone-sensitive lipase (HSL), (**c**) peroxisome proliferator-activated receptor γ (PPAR-γ), and (**d**) peroxisome proliferator-activated receptor-gamma coactivator (PGC-1α) mRNA levels. All values are presented as the mean ± standard error of the mean (*n* = 6). Differences between different groups were determined by a one-way ANOVA with Tukey test. Black dots represented each sample. Abbreviations: C, control group (ovariectomized (OVX) rats); L, low-dose group (OVX rats treated with 10 mg/kg body weight (BW) melatonin); M, medium-dose group (OVX rats treated with 20 mg/kg BW melatonin); H, high-dose group (OVX rats treated with 50 mg/kg BW melatonin).

**Table 1 nutrients-15-02800-t001:** List of primers sequences for the real-time qPCR.

Gene	Forward 5′→3′	Reverse 5′→3′
ACC	GGAAGACCTGGTCAAGAAGAAAAT	CACCAGATCCTTATTATTGT
FAS	AGCGGGAAAGTGTACCAGTG	GTAGCCGCAGCTCCTTGTAT
CPT-1	CACGAAGCCCTCAAACAGATC	CCATTCTTGAACCGGATGAAC
ACO	TGCAGACAGAGACGTAGGAAC	AAAGTGGTAGGCACGAATGC
PPARα	GCTCTGAACATTGGCGTTCG	TCAGTCTTGGCTCGCCTCTA
AMPK	ATGCCACTTTGCCTTCCGT	GCAGTTGCCTACCACCTCAT
LPL	ATGGCACAGTGGCTGAAAGT	CCGGCTTTCACTCGGATCTT
HSL	CGCGGACCAGCTCTAAAGAA	ATGATGGCACCTCCCTTTGG
PPARγ	CTGGGAGATCCTCCTGTTGAC	GGGAGTGGTCATCCATCACAG
FNDC5	AGGACAACGAGCCCAATAAC	CATATCTTGCTTCGGAGGAGAC
PGC-1α	TTCAGGAGCTGGATGGCTTG	GGGCAGCACACTCTATGTCA
Beta-actin	CACCAGTTCGCCATGGATGACGA	CCATCACACCCTGGTGCCTAGGGC

ACC, acetyl-CoA carboxylase; FAS, fatty acid synthase; CPT-1, carnitine palmitoyl transferase-1; ACO, acyl CoA oxidase; PPARα, peroxisome proliferator-activated receptor alpha; AMPK, 5′AMP-activated protein kinase; LPL, lipoprotein lipase; HSL, hormone-sensitive lipase; PPARγ, peroxisome proliferator-activated receptor gamma; FNDC5, fibronectin type III domain-containing protein 5; PGC-1α, peroxisome proliferator-activated receptor-gamma coactivator; Fw, forward; Rv, reverse.

**Table 2 nutrients-15-02800-t002:** Female hormone-related variables.

	C	L	M	H
Estradiol (pg/mL)	3262 ± 427.3	2430 ± 230.1	2614 ± 129.4	2778 ± 148.1
FSH (pg/mL)	6.95 ± 1.32	4.29 ± 0.71	8.35± 1.89	5.79 ± 0.50
Uterus weight (g)	0.22 ± 0.01	0.20 ± 0.03	0.17 ± 0.02	0.21 ± 0.02

Serum estradiol and follicular-stimulating hormone (FSH) levels were tested at 10 days after an ovariectomy. Uterus weights was tested after 8 weeks of the melatonin intervention. All values were presented as the mean ± standard error of the mean (*n* = 6). The differences between the different groups were determined by a one-way ANOVA with the Tukey’s test. Abbreviations: C, control group (ovariectomized (OVX) rats); L, low-dose group (OVX rats treated with 10 mg/kg body weight (BW) melatonin); M, medium-dose group (OVX rats treated with 20 mg/kg BW melatonin); H, high-dose group (OVX rats treated with 50 mg/kg BW melatonin).

**Table 3 nutrients-15-02800-t003:** Body weight (BW), BW gain, and food intake variables after 8 weeks of the melatonin intervention.

	C	L	M	H
Initial BW (g)	244.6 ± 2.72	240.6 ± 5.03	241.5 ± 5.12	244.7 ± 2.71
BW at 4 weeks after an OVX (g)	306.0 ± 9.39	302.3 ± 6.55	299.8 ± 9.75	306.4 ± 6.33
BW after 8 weeks of the melatonin intervention (g)	373.4 ± 9.98	347.0 ± 8.72	346.3 ± 12.68	353.3 ± 7.03
Total BW gain (g)	65.92 ± 5.05 ^a^	43.58 ± 4.45 ^b^	45.92 ± 3.21 ^b^	46.83 ± 2.84 ^b^
Food intake (g/day/rat)	23.49 ± 0.26	22.18 ± 0.74	21.52 ± 0.80	21.63 ± 0.52
Food efficiency (g BW gain/100 g diet)	5.38 ± 0.41 ^a^	3.75 ± 0.30 ^b^	4.08 ± 0.20 ^b^	4.18 ± 0.30 ^a,b^

All values are presented as the mean ± standard error of the mean (*n* = 6). Different superscripts (^a, b^) in a given row indicate a significant difference between different group at *p* < 0.05 by a one-way ANOVA with Tukey test. Abbreviations: C, control group (ovariectomized (OVX) rats); L, low-dose group (OVX rats treated with 10 mg/kg body weight (BW) melatonin); M, medium-dose group (OVX rats treated with 20 mg/kg BW melatonin); H, high-dose group (OVX rats treated with 50 mg/kg BW melatonin). Total BW gain was calculated by the following formula: ((BW after 8 weeks of the melatonin intervention) − (BW at 4 weeks after an OVX)); Food efficiency was calculated by the following formula: food efficiency = (total weight gain (g)/food intake (100 g diet)) × 100%.

**Table 4 nutrients-15-02800-t004:** Organ, adipose tissue, and muscle tissue masses after 8 weeks of the melatonin intervention.

	C	L	M	H
Liver weight (g)	10.88 ± 0.87	9.35 ± 0.49	9.03 ± 0.53	9.25 ± 0.28
Gonadal adipose tissue weight (g)	5.92 ± 0.34 ^a^	4.34 ± 0.35 ^b^	4.37 ± 0.43 ^b^	4.05 ± 0.21 ^b^
Perirenal adipose tissue weight (g)	7.39 ± 0.80 ^a^	4.32 ± 0.54 ^b^	5.19 ± 0.44 ^b^	4.45 ± 0.29 ^b^
Quadricep femoris muscle weight (g)	5.88 ± 0.25	5.39 ± 0.33	5.79 ± 0.30	5.84 ± 0.38
Gastrocnemius muscle weight (g)	4.78 ± 0.07	4.43 ± 0.09	4.54 ± 0.20	4.68 ± 0.11
Relative liver weight (g/100 g BW)	2.90 ± 0.16	2.69 ± 0.08	2.61 ± 0.10	2.62 ± 0.05
Relative gonadal adipose tissue weight (g/100 g BW)	1.58 ± 0.08 ^a^	1.25 ± 0.10 ^a,b^	1.41 ± 0.15 ^a,b^	1.15 ± 0.05 ^b^
Relative perirenal adipose tissue weight (g/100 g BW)	1.97 ± 0.18	1.55 ± 0.31	1.50 ± 0.12	1.27 ± 0.09
Relative quadricep femoris muscle weight (g/100 g BW)	1.58 ± 0.07	1.56 ± 0.10	1.67 ± 0.05	1.66 ± 0.11
Relative gastrocnemius femoris muscle weight (g/100 g BW)	1.29 ± 0.05	1.28 ± 0.04	1.32 ± 0.05	1.32 ± 0.02

All values are presented as the mean ± standard error of the mean (*n* = 6). Different superscripts (^a, b^) in a given row indicate a significant difference between different group at *p* < 0.05 by a one-way ANOVA with Tukey test. Abbreviations: C, control group (ovariectomized (OVX) rats); L, low-dose group (OVX rats treated with 10 mg/kg body weight (BW) melatonin); M, medium-dose group (OVX rats treated with 20 mg/kg BW melatonin); H, high-dose group (OVX rats treated with 50 mg/kg BW melatonin). Relative organ and tissue weights were calculated by applying the equation: relative organ weight = g/100 g BW.

**Table 5 nutrients-15-02800-t005:** Insulin resistance analysis after 8 weeks of the melatonin intervention.

	C	L	M	H
Insulin (µg/L)	0.21 ± 0.02	0.18 ± 0.01	0.18 ± 0.01	0.19 ± 0.03
Fasting glucose (mg/dL)	180.0 ± 32.83	137.2 ± 17.23	144.0 ± 9.78	138.3 ± 13.26
HOMA-IR	2.42 ± 0.58	1.45 ± 0.17	1.54 ± 0.10	1.62 ± 0.27

All values are presented as the mean ± standard error of the mean (*n* = 6). Differences between different groups were determined by a one-way ANOVA with Tukey test. Abbreviations: C, control group (ovariectomized (OVX) rats); L, low-dose group (OVX rats treated with 10 mg/kg body weight (BW) melatonin); M, medium-dose group (OVX rats treated with 20 mg/kg BW melatonin); H, high-dose group (OVX rats treated with 50 mg/kg BW melatonin); HOMA-IR, homeostasis model assessment of insulin resistance.

**Table 6 nutrients-15-02800-t006:** Semiquantitative lesion scores of rat’s livers.

Item	Definition	Score	C	L	M	H
Steatosis	<5%	0	0.00 ± 0.00	0.00 ± 0.00	0.00 ± 0.00	0.00 ± 0.00
	5–33%	1				
	>33–66%	2				
	>66%	3				
Inflammation	No foci	0	1.33 ± 0.58	0.33 ± 0.58	0.67 ± 0.58	1.33 ± 1.53
	<2 foci per 200× field	1				
	2–4 foci per 200× field	2				
	>4 foci per 200× field	3				
Necrosis	Normal (0%)	0	0.00 ± 0.00	0.00 ± 0.00	0.00 ± 0.00	0.00 ± 0.00
	Minimal (<1%)	1				
	Slight (1–25%)	2				
	Moderate (26–50%)	3				
	Moderately severe (51–75%)	4				
	Severe/high (76–100%)	5				
Total histological score			1.33 ± 0.58	0.33 ± 0.58	0.67 ± 0.58	1.33 ± 1.53

All values are presented as the mean ± standard error of mean (*n* = 3). Differences between different groups were determined by a one-way ANOVA with Tukey test. Abbreviations: C, control group (ovariectomized (OVX) rats); L, low-dose group (OVX rats treated with 10 mg/kg body weight (BW) melatonin); M, medium-dose group (OVX rats treated with 20 mg/kg BW melatonin); H, high-dose group (OVX rats treated with 50 mg/kg BW melatonin). The total histology score was calculated by summing the scores for steatosis, inflammation, and necrosis.

**Table 7 nutrients-15-02800-t007:** Semiquantitative lesion scores of rat’s adipose tissues.

Item	Definition	Score	C	L	M	H
Brite/beige cell	Normal (0%)	0	0.00 ± 0.00	1.00 ± 1.00	0.00 ± 0.00	0.33 ± 0.58
	Minimal (<1%)	1				
	Slight (1–25%)	2				
	Moderate (26–50%)	3				
	Moderately severe (51–75%)	4				
	Severe/high (76–100%)	5				

All values are presented as the mean ± standard error of the mean (*n* = 3). Abbreviations: C, control group (ovariectomized (OVX) rats); L, low-dose group (OVX rats treated with 10 mg/kg body weight (BW) melatonin); M, medium-dose group (OVX rats treated with 20 mg/kg BW melatonin); H, high-dose group (OVX rats treated with 50 mg/kg BW melatonin).

## Data Availability

The data of this study are available from the corresponding author upon reasonable request.

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
