# Peer review of "Effects of Melatonin Supplementation on Lipid Metabolism and Body Fat Accumulation in Ovariectomized Rats"

_nutrients, 2023, doi:10.3390/nu15122800_

Round 1
Reviewer 1 Report
The manuscript titled "Effects of Melatonin Supplementation on Lipid Metabolism and Body Fat Accumulation in Ovariectomized Rats" investigates the effects of melatonin supplementation on various variables in OVX rats. The study examines a wide range of Melatonin's effects, including Hormone-Related Variables, BW and Food Intake Variables, fasting glucose level and HOMA-IR index, Adipokine, Hepatic Lipid Metabolism, metabolism-related mRNA, histopathology, Irisin and Irisin-Related Levels, WAT Morphology, and Adipose Tissue Lipid Metabolism-Related Enzymes. The results indicate that OVX rats treated with melatonin supplementation exhibit lower body weights and body weight gain, particularly in the liver and gonadal or perirenal adipose tissues. However, no dose-dependent effect was observed. The authors further investigated lipid metabolism in the liver and found a significant reduction in ACC and FAS mRNA levels in the Melatonin groups. They also explored the mechanism behind the lower weight in the liver and adipose tissue and discovered that melatonin can reduce hepatic fatty acid synthesis and promote the browning of white adipose tissues through irisin. Overall, the manuscript's methods are clear and straightforward, and the results are comprehensive. The findings of this study could potentially provide useful insights for weight management.
Major Comments:
In terms of the investigation's scope, I have some concerns regarding the broad examination of RT-PCR and histopathological analyses in the liver, gonadal adipose tissue, and quadriceps. The study seems to encompass multiple genes and tissues, which may warrant further clarification. I recommend that the authors provide a clear rationale for their choices, explaining why specific genes were selected for analysis in the liver and adipose tissue, and how these choices align with the study's objectives. Providing this additional context would strengthen the manuscript and address concerns regarding the breadth of the investigation.
I found some of the results from the one-way ANOVA with Fisher's LSD test a bit confusing since the significance was not mentioned. For instance, in Figure 1a, the Control group appears to have a higher Leptin level compared to the L and H groups. In Figure 2a, the L group seems to exhibit a significant difference compared to the C group. It would be helpful to add dots to represent each sample and include p-values for the comparisons. Considering that you have 6 replicates in each condition and most of them are compared to the control group, I suggest considering additional statistical tests such as the Wilcoxon rank test or TukeyHSD after performing the one-way ANOVA.
Minor Comments:
In Table 2, please describe the significance of the results or add a sentence indicating that there was no significance among the groups. Additionally, I am curious about the p-value for the comparisons of Estradiol or FSH between the C and L groups.
For Table 3, Table 4, and Table 5, please clarify the meaning of the letters "a," "b," and "ab." I assume that different letters (a, b, c) in the same row indicate a significant difference at p < 0.05 determined using two-way ANOVA.
L345, please confirm if the statement "significant lower?"
The manuscript is well organized and quality of English is fine but needs slight improvements.
Author Response
Thanks for your excellent review and comments. We have incorporated the necessary changes in the revised manuscript point by point based on your comments. We have highlighted the changes in the original manuscript by using the red-color text.
- Major Comments:
Comment 1: In terms of the investigation's scope, I have some concerns regarding the broad examination of RT-PCR and histopathological analyses in the liver, gonadal adipose tissue, and quadriceps. The study seems to encompass multiple genes and tissues, which may warrant further clarification. I recommend that the authors provide a clear rationale for their choices, explaining why specific genes were selected for analysis in the liver and adipose tissue, and how these choices align with the study's objectives. Providing this additional context would strengthen the manuscript and address concerns regarding the breadth of the investigation.
Response : Thank you for your comments, we have added the information in the introduction to strengthen the relationship between postmenopausal obesity and lipid metabolic disorder in liver and adipose tissue (in L46-52) and the reason for selecting specific genes in the liver and adipose tissue (in L238-240, L301-303, L305-306).
Comment 2: I found some of the results from the one-way ANOVA with Fisher's LSD test a bit confusing since the significance was not mentioned. For instance, in Figure 1a, the Control group appears to have a higher Leptin level compared to the L and H groups. In Figure 2a, the L group seems to exhibit a significant difference compared to the C group. It would be helpful to add dots to represent each sample and include p-values for the comparisons.
Considering that you have 6 replicates in each condition and most of them are compared to the control group, I suggest considering additional statistical tests such as the Wilcoxon rank test or TukeyHSD after performing the one-way ANOVA.
Response : Thank you for your suggestion, we have added dots to represent each sample and the p-values for the comparison to the C group in figures 1,2,3,5, and 7. In addition, we have revised the statistical analysis method to the Tukey test and corrected the results (in Table3, Table 4, Table 5, and Figure 5).
- Minor Comments:
Comment 1: In Table 2, please describe the significance of the results or add a sentence indicating that there was no significance among the groups. Additionally, I am curious about the p-value for the comparisons of Estradiol or FSH between the C and L groups.
Response : Thank you for your reminder, and the p-value for the comparisons of Estradiol or FSH between the C and L groups are shown as follows.
|
|
p-value (C VS. L) |
|
Estradiol |
0.339 |
|
FSH |
0.389 |
Comment 2:For Table 3, Table 4, and Table 5, please clarify the meaning of the letters "a," "b," and "ab." I assume that different letters (a, b, c) in the same row indicate a significant difference at p < 0.05 determined using two-way ANOVA. "a," "b," and "ab.
Response : Thank you for your suggestion, we revised the footnote in Tables 3, 4, and 5.
Comment 3:L345, please confirm if the statement "significant lower?"
Response : Thank you for your reminder, we have revised in the L345 of the revised manuscript.
Reviewer 2 Report
The present work deals with an interesting topic. Introduction and Materials and Methods are, generally, clearly described. In the Results and Discussion, however, some criticisms should be highlighted.
Line 163: It is not clear how the ovariectomy success might be assessed in the absence of a not- OVX control group?
Line 182, Table 3: How was the BW gain calculated?
Lines 221,222: Are not the Authors surprised to have found, after 8 weeks of the melatonin supplementation, no significant differences in serum leptin or adiponectin levels among all groups, in spite of the adipose tissue weight decrease?
Line 262: Please, can the Authors explain which is the scoring system used for the liver and adipose tissue?
Line 278. Based on the Figure 5, the downward trend of the relative PGC-1α mRNA levels in the melatonin treated groups cannot be appreciated.
Figure 6. Based on the figure, it is difficult to recognize the presence of beige/brite adipocytes in the L group. A clearer figure is required or immunohistochemistry could be used to obtain a more convincing result.
Lines 344,345: …. hepatic fatty acid synthesis enzymes, ACC and FAS, were significantly higher in OVX rats treated with 50 mg/kg BW melatonin for 8 weeks. Compare with Results 3.6.2. and Figure 3
Lines 385, 386: Have you a comment to explain the failure of melatonin supplementation to affect browning in M group?
Line 393: a possible reason was FNDC5 had been cleaved into irisin and released into the blood circulation. Such an explanation can be referred to the protein, not to the FNDC5 mRNA levels. On the other hand, the authors found that there were no significant differences in the serum irisin levels among melatonin treatment groups.
Lines 401, 402: I think that this sentence, as well as the Conclusions, require more convincing findings to be accepted
Line 395; did not affected neither – to be corrected in: did affect (or affected) neither
Author Response
Please see attached file, thank you very much.
Comment 1: Line 163: It is not clear how the ovariectomy success might be assessed in the absence of a not- OVX control group?
Response: The previous study in our laboratory also used OVX rats as a menopausal animal model, which is the same experimental design in the present study[1]. Yeh et al. found that compared to sham-operated rats, OVX rats had significantly lower estradiol levels and uterine atrophy, accompanied by increasing BW gain and abdominal fat accumulation. In addition, the previous study showed that the serum estradiol level in the sham group was 57.52 ± 7.09 (ng/mL), and the weight of the uterus was 0.56 ± 0.08 (g), which were much higher than OVX rats in this experiment and there were no significant differences among all groups. Therefore, it is speculated that the ovariectomy in this experiment was successful.
|
Items |
In Yeh et al. study |
In this study |
|||
|
Sham group |
C |
L |
M |
H |
|
|
Estradiol |
5752 ± 709 |
3262 ± 427.3 |
2430 ± 230.1 |
2614 ± 129.4 |
2778 ± 148.1 |
|
Uterus weight (g) |
0.56 ± 0.08 |
0.22 ± 0.01 |
0.20 ± 0.03 |
0.17 ± 0.02 |
0.21 ± 0.02 |
Comment 2: Line 182, Table 3: How was the BW gain calculated?
Response: Thank you for your reminder, BW gain was calculated by the following formula:[ (BW after 8 weeks of the melatonin intervention)-(BW at 4 weeks after an OVX) ]. We have revised in the L175-176 of the revised manuscript.
Comment 3: Lines 221,222: Are not the Authors surprised to have found, after 8 weeks of the melatonin supplementation, no significant differences in serum leptin or adiponectin levels among all groups, in spite of the adipose tissue weight decrease?
Response :
Although there were no significant differences in serum leptin or adiponectin levels among all groups, compared to the C group, the Mel treatment groups showed a trend of lower serum leptin levels. Previous studies showed that melatonin reduced the serum leptin level [2,3], and another study indicated that the reduced leptin level after melatonin treatment was associated with reducing fat mass rather than improving leptin function[4]. The result of leptin level in this study was consistent with Szewczyk-Golec et al. study, as male rats treated with 20μg/ml Mel for 7 weeks had significantly reduced body weight, but there was no significant difference in leptin level[4]. Leptin and melatonin are secreted in a circadian rhythm with acrophases in the nocturnal period. Therefore, melatonin did not affect leptin level in this study, which might be related to the different time of sample collection, because of the rhythmic secretion of leptin.
However, Yeh et al. study pointed out that compared with the sham group, there was no significant difference in leptin level in OVX rats[1]. Consistent with Babaei et al. study, eight weeks after ovariectomy, the body weight and visceral fat mass were significantly increased in 11-week-old OVX rats but did not affect the adiponectin level. The possible reason is that OVX can promote adipocytes hyperplasia, not adipocyte hypertrophy, only adipocyte hypertrophy will reduce adiponectin level[5]. Therefore, we speculate that the OVX animal model in this study did not cause adiponectin secretion disorder. If the secretion is normal, the intervention of melatonin will not affect the adiponectin level.
Comment 4: Please, can the Authors explain which is the scoring system used for the liver and adipose tissue?
Response: Thank you for your reminder, we have added the information of histopathological evaluations in the L138-139, Table 6, and Table 7 of the revised manuscript.
Comment 5: Figure 6. Based on the figure, it is difficult to recognize the presence of beige/brite adipocytes in the L group. A clearer figure is required or immunohistochemistry could be used to obtain a more convincing result.
Response: In adipose tissue morphology, white adipocytes have one unilocular big lipid droplet, and Brite/beige adipocytes display a multilocular lipid droplet phenotype like the picture as follows.
In this study, as shown in Figure 6, the center in the pathological histologic images in the L group is the brite/beige adipocyte.
Comment 6: Lines 385, 386: Have you a comment to explain the failure of melatonin supplementation to affect browning in M group?
Response :
Thank you for your comment, we explain in the L384-393 of the revised manuscript.
Comment 7: Lines 344,345: …. hepatic fatty acid synthesis enzymes, ACC and FAS, were significantly higher in OVX rats treated with 50 mg/kg BW melatonin for 8 weeks. Compare with Results 3.6.2. and Figure 3
Response: Thank you for your reminder, we have revised in the L346 of the revised manuscript.
Comment 8: Line 393: a possible reason was FNDC5 had been cleaved into irisin and released into the blood circulation. Such an explanation can be referred to the protein, not to the FNDC5 mRNA levels. On the other hand, the authors found that there were no significant differences in the serum irisin levels among melatonin treatment groups.
Response: Thank you for your reminder, we explain in the L398-404 (Page 12)of the revised manuscript. Because of the change of statistical analysis method, the results of FNDC5 also changed.
Comment 9:Line 395; did not affected neither – to be corrected in: did affect (or affected) neither
Response: Thank you for your reminder, we have revised in the L406 (Page 12) of the revised manuscript.
